# Chromatin Structure and Drug Resistance in *Candida* spp.

**DOI:** 10.3390/jof6030121

**Published:** 2020-07-30

**Authors:** Callum J. O’Kane, Rachel Weild, Edel M. Hyland

**Affiliations:** School of Biological Sciences, Queen’s University Belfast, BT9 5DL Belfast, Northern Ireland; cokane44@qub.ac.uk (C.J.O.); rweild01@qub.ac.uk (R.W.)

**Keywords:** epigenetics, phenotypic resistance, histone acetylation, Candida, lysine deacetylase (KDAC) inhibitors

## Abstract

Anti-microbial resistance (AMR) is currently one of the most serious threats to global human health and, appropriately, research to tackle AMR garnishes significant investment and extensive attention from the scientific community. However, most of this effort focuses on antibiotics, and research into anti-fungal resistance (AFR) is vastly under-represented in comparison. Given the growing number of vulnerable, immunocompromised individuals, as well as the positive impact global warming has on fungal growth, there is an immediate urgency to tackle fungal disease, and the disturbing rise in AFR. Chromatin structure and gene expression regulation play pivotal roles in the adaptation of fungal species to anti-fungal stress, suggesting a potential therapeutic avenue to tackle AFR. In this review we discuss both the genetic and epigenetic mechanisms by which chromatin structure can dictate AFR mechanisms and will present evidence of how pathogenic yeast, specifically from the *Candida* genus, modify chromatin structure to promote survival in the presence of anti-fungal drugs. We also discuss the mechanisms by which anti-chromatin therapy, specifically lysine deacetylase inhibitors, influence the acquisition and phenotypic expression of AFR in *Candida* spp. and their potential as effective adjuvants to mitigate against AFR.

## 1. Candidiasis and the Prevalence of Anti-Fungal Resistance

Candidiasis, one of the most widespread human fungal infections, is caused by yeast species from the *Candida* genus. Over 30 *Candida* species have been isolated from infected patients, and the prevalence of each species varies, based on patient demographic and geographical location. However, over 90% of clinical isolates are typically accounted for by five species: *C. albicans, C. glabrata, C. tropicalis, C. parapsilosis,* and *C. krusei* [1,2,3]. These yeasts exist solely as commensals, which inhabit human skin and mucosal surfaces, under the restraint of our immune system. However, in immunocompromised hosts, virulent *Candida* species are opportunists and can overgrow, developing into a clinical infection. *Candida* infections are especially common in hospital settings and can lead to invasive mycoses with an associated mortality rate of ~40% [4,5,6].

Although *C. albicans* causes most candidiasis cases, non-*C. albicans Candida* [NCAC] infections have progressively increased in frequency over the past two decades [2,7]. Of these species, *C. glabrata* is the most frequently isolated NCAC species in global surveillance studies, representing 10-18% of all *Candida* isolates [8]. In 2009 a previously unseen *Candida* species was isolated from a patient in Asia [9], *Candida auris,* and has since spread rapidly across the globe to a total of six continents [10]. 

The global emergence of *C. auris,* and the increase in NCAC species can be partly attributed to improvements in diagnostic methods, but worryingly also from the lower treatment response rates of NCAC species [11]. Currently, there are only five main types of anti-fungal drugs; 1. Those that compromise the fungal cell wall, called echinocandins, which includes caspofungin, anidulafungin, and micafungin [12]. 2. Azole drugs, including fluconazole, itraconazole, voriconazole, and posaconazole which exert a fungistatic effect by inhibiting Erg11p, a lanosterol 14α-demethylase, essential for the biosynthesis of the fungal membrane lipid, ergosterol [13]. 3. The polyenes, including amphotericin B, which causes cell death by binding irreversibly to ergosterol in the cell membrane generating lethal membrane pores [14]. 4. The highly toxic inhibitor of nucleic acid biosynthesis, 5-flucytosine (5-FC) [15]. 5. The allylamines, including terbinafine, which inhibit the action of squalene epoxidase [16]. Certain *Candida* species are intrinsically resistant to particular anti-fungals, meaning that all isolates of this species show elevated minimum inhibitory concentrations (MIC) to a drug compared with other *Candida* species. (Here we will use MIC to refer to MIC_50_, which is defined as the lowest drug concentration that inhibits 50% fungal growth after 24 h). Examples of intrinsic resistance include azole resistance in *C. glabrata* and fluconazole resistance in both *C. krusei* and potentially *C. auris* [10,17,18]. For the most part, the molecular bases of intrinsic resistance in *Candida* spp. are unknown [19,20], although recent research suggests that in *C. auris* at least, transcriptional regulation is key [21].

The biggest threat these fungi pose to global health is the ease with which *Candida* species acquire anti-fungal resistance. Indeed, the recent isolation of multi-drug resistant *Candida auris* isolates in hospitals, as well as the acquisition of echinocandin resistance in the already azole-resistant *C. glabrata* [22,23,24], have propelled fungi into the medical spotlight. Therefore, with limited treatment options, and a rise in anti-fungal resistance, it is not surprising that the estimated mortality rate for certain human fungal diseases can be as high as 70–80% [5].

The acquisition and maintenance of drug resistance is an evolutionary process whereby fungi adapt to the presence of anti-fungals by acquiring genetic mutation [25,26]. Many previous studies have described distinct molecular pathways to resistance in *Candida* spp., and we are approaching an in-depth mechanistic understanding of the process [20,27,28,29]. The resistance mechanisms employed by *Candida* spp. towards specific drugs, include *de novo* mutations in anti-fungal targets, changes to cell wall/membrane integrity that limit anti-fungal uptake, expression of drug efflux pumps, and the modification and/or sub-cellular sequestration of an anti-fungal. However, the contribution of fungal chromatin structure and epigenetic mechanisms to anti-fungal adaptation and resistance is an under-explored aspect of this phenomenon, in spite of growing evidence supporting its therapeutic potential [30,31].

## 2. Chromatin Structure and Anti-Fungal Resistance

Fungi are eukaryotic organisms and as such their DNA is packaged inside nuclei in the form of chromatin, similar to higher eukaryotes. Fungal nuclei contain nucleosomes comprised of two copies each of the four core histone proteins, H3, H4, H2A and H2B wrapped around 147bp DNA. This organized compaction of DNA is not static but dynamically altered in response to the immediate needs of the fungal cell, by a group of proteins collectively called chromatin modifiers. These proteins include: (a) the enzymes that catalyze the post-translational modification of histone proteins, such as lysine (K) acetyl-transferases (KATs), and lysine (K) deacetylases (KDACs); (b) ATP-dependent chromatin remodelers which actively move nucleosomes in relation to the underlying DNA; and (c) histone variant proteins that are incorporated into non-canonical nucleosomes. There is an impressive body of work detailing the precise molecular consequences of many fungal chromatin modifiers [32,33,34]. However, in simplified terms, their collective action interferes with chromatin structure, regulating the accessibility of the DNA sequence for DNA-templated processes.

The acquisition and maintenance of AFR in *Candida* spp. is sensitive to chromatin structure and many studies have directly implicated various chromatin modifiers in the ability of *Candida* spp. to survive in the presence of anti-fungal drugs. In *C. albicans,* the deletion of each of the KDACs *Ca_RPD31,* and *Ca_HDA1* sensitizes the yeast to fluconazole, itraconazole and voriconazole [35]. Consistently, the removal of the KAT genes *Ca_HAT1* and *Ca_HAT2* achieves the reverse, increasing *C. albican’s* azole resistance [36]. Furthermore, although loss of function of the KAT *Ca_RTT109* has no effect on azole resistance, the homozygous deletion of this locus decreases the ability of *C. albicans* to withstand both echinocandins and 5-FC [37]. In contrast, *C. albicans* strains lacking the sirtuin KDACs, *Ca_HST3 [Ca_HST3+/Δ] *and *Ca_HST1 [Ca_hst1Δ/Δ] *showed either no effect or increased resistance to azoles respectively [37,38]. In 2009, Sellam et al. showed that the deletion of the ADA2 component of the multi-subunit Spt-Ada-Gcn5-acetyltransferase [SAGA] co-activator complex in *C. albicans,* leads to a 90% decrease in survival in the presence of high concentrations of fluconazole compared with the non-mutated strain [39]. This is in contrast to the phenotypic characterization of the *Ca_gcn5Δ/Δ* deletion strain, the catalytic subunit of the same SAGA complex, which displayed unaltered azole susceptibility but hyper-sensitivity to caspofungin [40]. Taken together these studies illustrate how resistance mechanisms in *C. albicans* at least are dependent on the function of specific chromatin modifiers, most notably those that regulate histone acetylation.

What are the underlying mechanisms by which chromatin modifiers impact AFR? Theoretically, there are three ways by which the regulated compaction of DNA into chromatin may direct the path to anti-fungal resistance and/or maintain resistance mechanisms (summarized in Figure 1).

1.Chromatin structure can influence the availability of particular anti-fungal resistance mutations, and thus contribute to the genetics of AFR.2.Chromatin modifiers can impact epigenetic mechanisms that can give rise to fungal phenotypic plasticity and “phenotypic resistance”.3.Transcriptional regulation by chromatin structure can determine the level of expression of anti-fungal resistant genes, contributing to the phenotypic expression of AFR.

In this review we will focus our discussion on possibilities 1 and 2, and direct readers to other reviews that describe the transcriptional regulation of anti-fungal resistance mechanisms [41,42]. 

## 3. Chromatin Structure and the Availability of Anti-Fungal Resistance Mutations

### 3.1. Beneficial Mutations and Evolution 

Darwinian adaptation relies on the selection of pre-existing beneficial mutations that promotes reproductive success of an organism. In the case of anti-fungal resistance, beneficial mutations can be simply defined as genomic alterations that counteract and/or minimize the effect of an anti-fungal, allowing the yeast to grow and divide in its presence. However, in reality the success of a given beneficial mutation is also impacted by evolutionary factors [43,44] and population dynamics [45,46], such as fitness tradeoffs, population size, the strength of selective pressure and clonal interference, to name a few. Together with the fact that genetic mutation can be a random event, it is not surprising that the exact genetic mutation that gives rise to anti-fungal resistance can vary between species, between populations and even between individual isolates. This is no more evidenced than when comparing azole resistance mechanisms in *C. albicans* and *C. glabrata.* While the majority of *C. albicans* azole resistant isolates contain a mutation in the binding site of the drug target, Ca_Erg11p, large-scale studies on resistant *C. glabrata* isolates have revealed that mutations affecting either the expression levels or gene product of *Cg_ERG11* seldom occur [47,48,49,50]. By far the preferred mechanism of azole resistance in *C. glabrata* is the acquisition of mutations in the *Cg_PDR1* transcription factor gene that regulates the expression of the Cg_Cdr1p drug efflux pump [47,50,51]. The underlying molecular events that dictate this interspecies mutational bias in azole resistance remain unclear, but potentially evolutionary parameters, such as differences in fitness trade-off between the individual resistance mutations in each species, may account for these observations [52].

### 3.2. Chromatin Structure and Mutation Rate

Another variable that dictates the mutational landscape of a genome is chromatin structure. It is acknowledged extensively in humans that the 3D arrangement of DNA can dictate the accessibility of specific genomic aberrations [53]. For example, there is a positive correlation between somatic mutation rate and densely packed DNA in human tumor cells [54]. In *S. cerevisiae,* multiple studies report an impact of chromosome location on acquired genetic variation. It is known that: (i) mutation rate is not constant across the length of a *Saccharomyces cerevisiae* chromosome [55] and heterochromatic regions have a higher per base substitution rate [56]; (ii) there is a 10-15% reported lower substitution rate in inter-nucleosomal linker-DNA compared with nucleosomal DNA [57]; and (iii) changes to chromatin associated proteins change the vulnerability of *S. cerevisiae* to different types of genetic mutation and variation [58,59]. The underlying mechanism(s#) to explain these observations in yeast can be fundamentally explained by differences in the accessibility of certain DNA sequences dictated by nucleosome position. However, these differences in DNA accessibility impact other aspects of DNA metabolism such as DNA repair efficiencies, replication timing and gene expression, all of which have been linked to mutation rate [60,61].

For a yeast cell facing the stress of a fungicidal agent, the availability of precise mutations that alleviate this stress might be dictated at the level of chromatin structure. For example, the gene encoding a drug target that is positioned in a genomic location with decreased mutation rate, has a lower probability of gaining a resistance mutation. It remains to be examined whether the aforementioned bias in azole resistant alleles between *C. glabrata* and *C. albicans* is influenced by the local chromosomal context of the relevant resistance genes in either/both species.

### 3.3. Chromatin Structure and Genome Stability

Secondly, chromatin structure can play an indirect role in the availability of AFR mutations, in a mechanism analogous to ‘driver’ mutations in cancer cells. Driver mutations predispose normal cells to tumorigenesis, primarily, but not exclusively, by weakening genetic stabilization mechanisms. Mutations in human chromatin associated factors have been identified as driver mutations, which lead to both an increase in tumor cell mutation rate, as well as giving rise to favorable gene expression profiles [62]. In an analogous way, mutations in fungal genes that dictate chromatin function may similarly destabilize the fungal genome and facilitate the acquisition of further beneficial alleles that promote anti-fungal resistance. Consistently, it has been shown in *S. cerevisiae,* and indeed *C. albicans,* that the machinery regulating the chromatin modification, histone H3 lysine 56 acetylation (H3K56ac) is involved in promoting genome stability [63,64]. For example, when the H3K56-specific lysine deacetylases (KDACs) *HST3* and *HST4* are simultaneously deleted in *S. cerevisiae* (*hst3Δhst4Δ*), mutation avoidance mechanisms are impaired and mutation rate elevates to 27 times that of isogenic wild-type strains [65]. Although direct mutation rates have not been determined in homologous *Candida* deletion stains, numerous publications show that *C. albicans* lacking H3K56ac machinery is more susceptible to genotoxic agents such as hydroxyurea, methyl-methanesulfonate and camptothecin, indicating that this histone modification also plays a role in genome stability and DNA repair mechanisms in a pathogenic fungus [37,66]. Given that the homozygous deletion of the *C. albicans* H3K56ac specific lysine acetyl-transferase *Ca_rtt109Δ/Δ* sensitizes this species to the echinocandins and 5-FC anti-fungals, it suggests an unexplored link between histone H3K56ac, genome stability and AFR in *C. albicans* [37].

Interestingly, it has been shown in *C. albicans* that genome stability is altered by environmental triggers to promote the microevolution and adaptation of this species. For example, two recent studies found that *C. albicans* mutation rate is elevated during host infection [67], and in the presence of anti-fungal drugs [68] presumably to provide genetic raw material to the organism for survival under these stresses. Both studies also reveal the inconsistency of acquired genetic variation and how genome characteristics, such as differences in ploidy [68] and sub-telomeric chromatin structures [67], disproportionately elevate mutation mate. The mechanism by which such stress-induced mutagenesis arises remains to be determined, but taken together these findings suggest that regulators of genome architecture and chromatin structure may play a role in this process. Indeed heterochromatin, or densely packed chromatin structure, is recognized as a facilitator of genome plasticity in multiple fungal species [69]. One very interesting question would be whether H3K56ac levels are altered during stress induced mutagenesis in *Candida* spp.

## 4. *Candida* spp. Phenotypic Plasticity and Anti-Fungal Resistance

Phenotypic plasticity is primarily the relaxation of the barriers that restrict gene expression allowing organisms to express different phenotypes in response to environmental changes [70]. However, we acknowledge that broader definitions of this term do exist as outlined in the review by Kelly et al. [71]. Undisputed however, is that phenotypic plasticity is a highly advantageous evolutionary strategy that can facilitate rapid adaptation to novel environments [72]. For unicellular organisms, including pathogenic yeast, phenotypic plasticity manifests as population heterogeneity, whereby subpopulations of cells express alternate transcriptional states leading to changes in cellular metabolism, physiology, and ultimately phenotype.

### 4.1. Epigenetic Inheritance of Phenotypic Plasticity

The genetic architecture of phenotypic plasticity is not fully known. However, the majority of evidence to date suggests that epigenetic and not genetic mechanisms dictate phenotypic plasticity in pathogenic microbes [73,74]. Epigenetics is defined as the inheritance of a phenotypic state that is not dependent on DNA sequence [75]. Indeed, the different morphology switch phenotypes (see below) of *C. albicans and* C. *tropicalis*, once established, are stably inherited over several generations, implying epigenetic inheritance [76]. An epigenetic based mechanism allows organisms to avoid commitment to permanent genetic mutation, ensuring that phenotypic changes are reversible, a consequence that is evolutionarily beneficial if an environment reverts to its previous or an alternate state. In S*accharomycotina* yeast species, including *Candida* spp., the primary mode of epigenetic inheritance is through the post-translational modification of histone proteins that comprise the nucleosome [77]. However, in other fungal species, RNA interference (RNAi) mechanisms play an epigenetic role [78], but key players in the RNAi machinery are absent from the genomes of *Candida* spp., [79] and it is currently believed that these species do not avail of RNAi-based epigenetic inheritance.

Research over a number of years has demonstrated a clear link between *Candida* spp. phenotypic plasticity and AFR [80]. There are multiple phenotypic states in *Candida* spp. that alter anti-fungal susceptibility and give rise to phenomena such as biofilm growth, morphological epigenetic switching, anti-fungal tolerance and fungal persister cells. [81,82,83]. Such ‘phenotypic resistance’ is distinct from AFR in that the cells’ ability to withstand anti-fungal drugs is not genetically defined but relies on changes at transcriptional and post-transcriptional levels. Phenotypic resistance is a consequence of phenotypic heterogeneity within a given strain, and is, by definition, epigenetically inherited. For the purpose of this review, phenotypic resistance will broadly refer to non-genetic based mechanisms that facilitate survival of a fungal pathogen in the presence of anti-fungals at concentrations above the MIC, analogous to a similar phenomenon described for the bacterial response to antibiotics [84].

One evolutionary advantage to phenotypic resistance is that, theoretically, it can ‘buy time’, which would provide the cell with an opportunity to acquire genetic lesions that confer true resistance, thus maximizing an organism’s survival. Indeed, the adaptation to anti-fungal drugs might be initially attributed to a phenotypic switch to a less drug-susceptible state, which subsequently leads to acquired resistance (Figure 2) [85]. Although experimental evidence is currently lacking to support this hypothesis in *Candida* spp., such a combined epigenetic and genetic model of acquired drug resistance is clearly evident in tumor cells [86].

Broadly speaking, the underlying mechanisms that confer phenotypic resistance are not very well understood. Limited evidence suggests that phenotypic states with increased anti-fungal resistance are due to low intracellular anti-fungal accumulation, either through mechanisms that enhance drug efflux or reduce drug uptake [see below for specific examples]. Indeed, phenotypic resistance may depend on similar mechanisms as acquired resistance, but they are not genetically hard-wired and instead result from distinct transcript profiles. Other evidence suggests that cells which are phenotypically less susceptible to anti-fungals are also less metabolically active, as a decreased growth rate makes them resistant to the action of anti-fungal drugs that rely on cell growth for their effects. This is especially true for fungal persister cells, which are thought to be in an inactive dormant state [87]. Interestingly, slower growing cells within a yeast population are more prone to DNA damage than their fast-growing counterparts, suggesting an increased potential of acquiring anti-fungal resistance mutations [88], supporting the model presented in Figure 2.

The most direct way to identify the molecular underpinnings of phenotypic resistance is through single-cell analysis, in order to characterize population heterogeneity at the level of metabolism, physiology, and transcription [89]. One such study has employed DROPseq technology [90] to monitor the single-cell transcriptome of *C. albicans* when exposed to anti-fungal drugs at concentrations below the MIC [85]. These authors found that the population divided into two distinct groups after 48 h, due primarily to mounting different expression profiles of stress response genes. Interestingly this bimodal transcriptional response occurred in the presence of either fluconazole or caspofungin indicating that the mode of action of specific anti-fungal drugs does not influence the outcome. These data suggest that, upon anti-fungal stress, *C. albicans* cells explore different epigenetic landscapes, potentially to maximize survival. Furthermore, the authors found that a sub-population of cells acquired resistance after 72 h, although it was undetermined whether this was due to genetic, or possibly epigenetic, alterations [85].

Given the implied epigenetic inheritance of phenotypic resistance, it is imperative to understand the role of chromatin dynamics and modification in the acquisition and maintenance of such drug tolerant states. To date it is known that chromatin modification, most notably histone acetylation, is required for biofilm growth, morphology switching and anti-fungal tolerance. However, the field is still in the early stages and much remains to be discovered. We will discuss each of these states in turn and present evidence that support a role for epigenetic mechanisms in phenotypic resistance.

### 4.2. Candida Biofilms and Phenotypic Resistance

One of the first lines of evidence that phenotypic plasticity influences AFR was the discovery that fungal cells grown in a biofilm are recalcitrant to anti-fungal therapy [91,92]. Biofilms exploit the fact that fungal species, including most *Candida* spp., are dimorphic and can under particular conditions switch between two growth morphologies, yeast (single-celled) growth and hyphal (multicellular/multinucleate) growth [93,94], the so-called yeast-to-hyphal (Y-H) transition. Most *Candida* biofilms are complex communities of these two types of fungal cells, as well as secreted extracellular matrix components, which form on both biotic and abiotic surfaces [95]. Biofilm phenotypic resistance is due to numerous mechanisms including, but not limited to, drug sequestration by the biofilm matrix [96] and up-regulation of drug efflux pumps, [97] both of which result in lower intracellular concentrations of the anti-fungal drug. Furthermore, evidence exists that *Candida* spp. biofilms contain metabolically inactive persister cells that have increased anti-fungal tolerance to fungicidal drugs [87,98]. The idea has long been described for bacteria, where persister cells are a phenotypically distinct population of cells that are resistant to antibiotics, despite being genetically identical to their antibiotic susceptible counterparts [99,100]. However, for *C. albicans* at least, this assertion is currently under dispute [101,102]. Taken together, it is therefore not surprising that *Candida* biofilms are thought to provide a microenvironment that promotes the emergence of multi-drug resistant infections [103,104,105].

### 4.3. Histone Acetylation and Dimorphic Growth

There are many documented links between histone acetylation and phenotypic plasticity in *Candida* spp. [76]. During the transition from planktonic to biofilm growth for example, the rewiring of the *C. albicans* transcriptome is dependent on the function of the histone deacetylase complex SET3C, which modulates transient expression changes of key morphogenesis-related genes [106,107]. Interestingly, when this SET3C complex is compromised (*Ca_set3Δ/Δ* and *Ca_hos2Δ/Δ*), yeast to hyphal morphogenesis in *C. albicans* is surprisingly stimulated, not inhibited, and these mutant strains produce wrinkled colonies on rich media [106,108]. Consistently, *C. albicans* biofilms formed by cells lacking SET3C function have altered physical characteristics compared to WT cells, including increased biomass, enhanced cohesiveness and, most importantly, increased phenotypic resistance [109].

Other histone deacetylase*s* have been implicated in the Y-H transition in *C. albicans*. In contrast to the *Ca_set3Δ/Δ phenotypes,* both Zacchi et al. and Lu et al. have shown that *C. albicans* lacking the *Ca_HDA1* KDAC gene struggle to maintain hyphal growth on solid media [108,110], suggesting that this chromatin modifier is an activator of the Y-H transition. The KDAC, *Ca_RPD31,* has dual functionality in regulating hyphal morphogenesis in *C. albicans*. Phenotypic analysis of *Ca_rpd31Δ/Δ* shows that it does not form hyphae under conditions that stimulate Y-H transition [111]. However, under non-hyphal inducing conditions, and when combined with a mutation in the transcription factor *Ca_SSN6 (Ca_ssn6Δ/Δ),* lack of *Ca_RPD31* activity promote*s* hyphal development. These data indicate that *Ca_RPD31* functions as both an activator and a repressor of hyphal growth depending on environmental cues. Lastly, the sirtuin KDAC, *Ca_SIR2,* negatively regulates the Y-H transition as cells lacking this gene will spontaneously switch to hyphal morphology under non-inducing conditions [112].

The sensitivity of Y-H transition to histone acetylation is also evident in mutant strains with compromised KAT functionality. In *C. albicans* strains lacking a subunit of the NuA4 KAT complex *(Ca_yng2Δ/Δ*), or those containing the mutation *Ca_esa1*Δ/Δ, hyphal development on both solid and liquid media is severely affected [113,114]. Similarly, loss of function of the *Ca_Ada2p,* which makes up part of the SAGA KAT complex renders *C. albicans* unable to produce hyphal filaments in vitro, and interestingly, also in vivo in a *Caenorhabditis. elegans* model of candidiasis [115]. Consistently, deletion of the catalytic subunit of SAGA (*Ca_gcn5Δ/Δ)* leads to a severe morphogenesis defect and lack of hyphal formation [40,116]. Histone H3K56ac is regulated by the KAT and KDAC *Ca_HST3* and *Ca_RTT1109,* respectively. Loss of function of either of these genes leads to constitutive hyphal formation, indicating the sensitivity of morphogenesis to regulated H3K56ac levels [37,66].

Given that phenotypically resistant biofilm communities depend on *Candida* spp. dimorphism, there is a compelling argument that biofilm resistance can be altered through the manipulation of the histone de/acetylation pathways involved in Y-H transition. However, the data currently are not sufficient to identify the optimal target. For one, the effect of the majority of these KAT and KDAC mutants on biofilm development, biofilm robustness or indeed biofilm-related anti-fungal susceptibility is unknown. Furthermore, given the reliance of biofilm phenotypic resistance on extracellular matrix production [96], and potentially persister cells, it would be interesting the assess any matrix-related phenotypes, or alterations in fungicidal persistence in the histone modifier mutant strains.

Most importantly, it is not known how well these results in *C. albicans* can be extended to histone acetylation regulation in NCAC species. Additionally, data using both the *Ca_SET3* and *Ca_RPD31* deletion mutants, as well as strains lacking the regulators of H3K56ac, reveal significant complexity in how these pathways operate that can depend on factors such as timing, environmental cues, and the finely tuned regulation of specific chromatin modifications, respectively. Therefore, we also need to know when and how to perturb these histone acetylation pathways for maximum effect on biofilm phenotypic resistance.

### 4.4. Phenotypic Switching and Histone Acetylation

Certain *Candida* species undergo stochastic phenotypic switching in a small number of cells within a population, including *C. albicans* [117], *C. glabrata* [118], *C. lusitaniae* [119], and *C. tropicalis* [120]. These events occur at a high frequency, in some cases exceeding the rate of somatic mutation, are reversible, and represent distinctive epigenetic states of genetically identical cells. Although for each species the trigger and phenotypic outcomes of such spontaneous switching differ, they may all serve as bet-hedging strategies to maximize the success of each fungi under different environmental conditions, as is observed in *S. cerevisiae* [121] and bacteria [122]. For commensals and opportunistic strains, switching phenotypes can serve to maximize the colonization of their human host [123,124] including strategies such as host immune avoidance [125,126].

Perhaps not surprisingly, for certain fungal species where switching occurs, one or more of the cell types involved display anti-fungal phenotypic resistance. Of the five distinct switching phenotypes associated with C*. tropicalis*, the ‘crepe’ variant displays increased tolerance to itraconazole, whereas the ‘rough’ variant was less resistant to itraconazole [127]. Differences in anti-fungal susceptibility are also evident for distinct *C. lusitaniae* variants [119,128] as well as in *C. albicans* strains with a higher ‘white-opaque’ (W-O) switch frequency [129]. Furthermore, a case study has shown that W-O switching in *C. albicans* correlates with changes in anti-fungal susceptibility during successive Candida infections [130]. Therefore, high frequency switching is another example of how phenotypic plasticity can benefit the survival of pathogenic fungi when exposed to anti-fungal drugs.

Many papers have focused on the role of epigenetic mechanisms in W-O switching in *C. albicans.* Interestingly, the majority of the *C. albicans* mutant histone acetylation strains with defects in Y-H transition also show altered rates of W-O switching. In 2009, Hnisz et al. demonstrated that the following histone acetylation-related genes positively regulate *C. albicans* W-O switching; the sirtuin *Ca_HST2,* components of the SET3C complex, *Ca_SET3* and *Ca_HOS2*, and the KAT *Ca_NAT4* [131]. Interestingly, the C*a_nat4Δ/Δ* mutation also destabilized the opaque phenotype, with a three-fold increase in the rate of O-W switching, further supporting its role in maintaining the opaque epigenetic state [131]. W-O switching is also sensitive to H3K56 acetylation as the deletion of *Ca_RTT109* and *Ca_HST3* give rise to opposing switch phenotypes. *Ca_rtt109Δ/Δ* decreases W-O switching rate, whereas the heterozygous mutant *Ca_HST3+/Δ* increases it [132]. Furthermore, loss of *Ca_RTT109* destabilizes the opaque state, as growing *Ca_rtt109Δ/Δ* opaque colonies for 24 h in liquid media leads to a loss of up to 90% of cells expressing opaque-cell markers [132]. Similar to what was seen for Y-H transition, the removal of the *Ca_RPD31* gene impacted switching phenotype in both directions suggesting that Ca_Rpd31p acts to suppress both the W-O and O-W switch [133]. 

The deletion of other regulators of histone acetylation gave rise to phenotypes that are consistent with functioning as W-O switch repressors, displaying an increase in W-O switching frequency. These include the KDACs *Ca_HDA1*, [133] *Ca_SIR2* [112] and *Ca_HST3* [132], as well as the KATs *Ca_HAT1*, *Ca_HAT2* [134] and a subunit of the NuA4 acetylation complex *Ca_YNG2* [135]. Taken together these data indicate that inhibiting the function of these lysine acetyl transferases will give rise to a higher proportion of the *C. albicans* opaque state within a population.

Although these data clearly indicate that chromatin modifiers are required for *C. albicans* W-O switching, many of the detailed mechanistic studies have yet to be undertaken. For example, it is known that propagation of the opaque cell phenotype is dependent on an autonomous transcriptional feedback loop, involving the master regulator Ca_Wor1p [136], yet only limited information exists on how each of the aforementioned histone de/acetylation genes, or chromatin structure in general, feeds into this epigenetic-based mechanism [107,131]. Moreover, currently no mechanistic links have been drawn between histone acetylation and phenotypic switching in either *C. glabrata*, *C. lusitaniae*, or *C. tropicalis*, although undoubtedly chromatin regulation plays a role. 

AFR is not thought to be the primary benefit of phenotypic switching and there is a dearth of evidence suggesting that switching rates increase in the presence of anti-fungal drugs. However, the added advantage that specific switch variants of *Candida* spp. show altered anti-fungal susceptibility, is of clinical significance especially given that switching frequencies in *C. albicans* for example, are higher in pathogenic strains compared to commensals [137], and in strains that cause invasive candidiasis versus those causing superficial infections [138]. Furthermore, it is known that the *C. albicans* opaque cell phenotype is metabolically less active than the default white state under a variety of growth conditions, including under stress [139,140]. Therefore, switching to an opaque state could potentially attenuate an anti-fungal that relies on cell growth for efficacy. Given that perturbation of histone acetylation can block phenotypic switching, pharmaceutical intervention of the process could remove another avenue for the fungal cell to survive in the presence of anti-fungal drugs.

### 4.5. C. albicans Trailing Phenotype and Epigenetics

More recently it has been discovered that *Candida* species can generate sub-populations of specialized anti-fungal tolerant cells. Such tolerant cells are thought to explain the so-called ‘trailing’ phenotype exhibited experimentally at very high concentrations (above MIC) of fungistatic drugs, whereby low level of persistent cell growth is detected after 48 h of incubation under these conditions [81,82]. Such trialing growth has been described for multiple *Candida* species [141] and is detected in up to 60% of *C. albicans* clinical isolates [142]. Furthermore, AFR measured by MIC, and tolerance measured by the extent of trailing growth of *C. albicans* isolates do not correlate, consistent with the idea that they are independent responses to anti-fungal stress [143]. Although the clinical relevance of the trailing phenotype is under debate [144], such anti-fungal tolerant cells are readily isolated from patients with persistent candidemia [143] and those with prolonged *C. albicans* oral carriage [145]. These observations suggest that the ability of tolerant cells to grow in the presence of anti-fungals contributes to recurrent *Candida* infections.

Similar to other phenotypic resistant states, evidence suggests that *Candida* tolerance is mediated by the up-regulation of efflux pumps in tolerant cells [143], or potentially an increase in cell wall chitin in tolerant cells, which limits drug uptake [146]. A more detailed description of associated molecular mechanisms can be found in a recent review on the topic [81]. The contribution of histone acetylation to trailing growth has been investigated solely through the use of histone lysine deacetylase inhibitors (KDACIs). These potential adjuvant anti-fungal drugs will be discussed in detail in Section 5, but the data reported thus far indicates that the broad spectrum KDACI, Trichostatin A (TSA) and uracil-based hydroxamic acids can effectively reduce, and in some instances eliminate, fluconazole trailing growth in *C. albicans* clinical isolates [35,147,148]. Furthermore, TSA also works synergistically with itraconazole to limit trailing growth in both *C. albicans* and *C. tropicalis* [147]. By preventing the expression of tolerance, KDACIs might succeed in limiting the emergence of AFR consistent with the model presented in Figure 2**.** However, this remains to be fully investigated, and specifically whether KDACIs can influence therapeutic outcomes (see below).

Although anti-fungal tolerance and trialing growth is epigenetically defined, genetic variation between isolates of a given species impact strain specific differences, such as size of a tolerant sub-population and the degree of tolerance. Additionally, the extent to which tolerance is a reproducible and evolvable phenomenon suggests that genetic contribution(s) play a central role. Furthermore, although KDACIs could potentially be used to inhibit the emergence of acquired anti-fungal resistant clones, similar to what has been seen in using epigenetic-based anti-cancer therapy [86], this effect may depend on the starting epigenetic state of a given isolate, which paradoxically may reflect its genomic sequence. Realistically, the anti-fungal drug tolerant state is multifactorial and as such, improved experimental design, methods and data interpretation, using guidelines set out by Berman and Krysan [81], will help delineate between the epigenetic and genetic contributions.

## 5. KDAC Inhibitors: Pharmacological Modulation of Anti-Fungal Resistance in *Candida* spp.

As evident from the data in our previous sections, histone acetylation, above all other pathways that regulate chromatin structure, has the strongest established link to *Candida* spp. AFR. Many of the genetic studies are consistent with the simple model depicted in Figure 3 whereby the deletion of histone acetylation genes promotes *Candida* spp. AFR, whereas genetic inhibition of KDAC function, attenuates it. Therefore, it is no surprise that the therapeutic potential of KDACIs to combat fungal disease has been explored [31,149]. KDACIs as a mono therapy have limited effect on overall fungal viability both in vitro and in vivo, *for Candida* spp. [147,148]. However, consistent with the aforementioned genetic data, KDACI treatment of *Candida* spp. can impair phenotypic plasticity, alter stress response pathways and chromatin structure, and as such can interfere with the yeast’s ability to tolerate and/or respond to anti-fungal drugs. Consistently, a growing number of studies highlight the synergistic anti-Candida effect of KDACIs with common anti-fungals.

At this point it should be noted that, contrary to the simplified model above, the lysine acetyl-transferase (KAT) inhibitor, cyclopentylidene-(4-(4-chlorophenyl) thiazol-2-yl) hydrazone (CPTH2) has fungistatic properties by itself in vitro against *Candida* spp. from the CTG-clade [147]. Currently it is unclear what the specific fungal molecular target of this compound is, but genetic evidence suggests it is not Gcn5p [150]. Interestingly CPTH2 is more effective towards caspofungin resistant *Candida* isolates [150], suggesting not only a mechanistic link between echinocandin resistance and the CPTH2 target, but also its potential therapeutic value in treating problematic fungal infections. 

In fungi there are three distinct classes of fungal KDAC proteins, (classes I, II, III) totaling 10+ individual genes for majority of the sequenced *Candida* species [149]. KDACIs can be described as pan-inhibitors that target multiple KDAC classes, which are class-selective and target a single class, and even isoform selective thus distinguishing between proteins of the same class. Historically the majority of these small molecules have been designed to target human KDAC proteins for use in cancer treatment [151]. However, there is enough sequence conservation in the KDAC protein repertoire between humans and yeast that many ‘human specific’ KDACIs are also effective in fungal cells.

### 5.1. Class I and Class II KDAC Inhibition in Candida spp.

One of the first KDACIs analyzed for fungal disease was Trichostatin A (TSA). TSA was originally extracted from the metabolites of gram-positive *Streptomyces hygroscopicus*, and described as a fungistatic drug, acting against *Aspergillus niger* and *Trichophyton* spp. [152]. Over a decade later, TSA was found to inhibit mammalian histone deacetylases and primarily came under investigation alongside later developed KDACI drugs as a potential anti-cancer and anti-inflammatory agent [153,154,155]. TSA is a broad spectrum KDACI and acts as a non-selective inhibitor of the zinc-dependent Class I and II KDACs. In *C. albicans,* treatment with TSA has a dramatic effect on phenotypic plasticity and stimulates the Y-H conversion [106] as well as increases the rate of WO switching [156]. Furthermore, in the presence of TSA, *C. albicans* fluconazole trailing is reduced >200 fold [35,147], thereby increasing the effectiveness of azole treatment, potentially by either eliminating or preventing the generation of drug tolerant cells. Consistent with the model in Figure 2**,** TSA can also inhibit the emergence of azole resistance in *C. albicans* [25]. Synergistic effects were also noted between TSA and itraconazole for *C albicans*, as well as *C. parapsilosis* and *C. tropicalis* trailing growth [147], suggesting that KDAC inhibition combined with anti-fungals could potentially be a therapeutic option for NCAC infections.

Another type of Class I/II KDACI are those that contain uracil, the prototype being suberoylanilide hydroxamic (SAHA), which is sold under the name vorinostat [157]. In a 2007 study, four highly potent uracil-based hydroxamic acid KDACIs, as well as SAHA, were tested on both *C. albicans* and *C. parapsilosis* [148]. Unlike TSA, limited synergism was detected with these KDACIs and fluconazole, except against one particular *C. albicans* isolate which showed reduced fluconazole trailing in the presence of two of the KDACIs under investigation. However, the study did show that the same two uracil-based KDACIs successfully prevented the acquisition of fluconazole resistance in *C. albicans* under experimental conditions, supporting a role for histone acetylation during the immediate adaptation of *C. albicans* to anti-fungal drugs, in line with the model presented in Figure 2.

The short-chain aliphatic acid, sodium butyrate, is a KDACI whose action predominantly interferes with class I KDACs. Sodium butyrate is one of the few KDACIs that by itself can negatively impact *C. albicans*, as well as *C. parapsilosis* growth [158], although this is not a consistent finding [147]. Furthermore, sodium butyrate reduces biofilm formation in both *C. albicans* and *C. parapsilosis,* as well as sensitizing these species to fluconazole [147,158]. Interestingly, sodium butyrate can enhance the activity of amphotericin B against *C. albicans* biofilms [159]. In *C. albicans* there are four Class I KDACs, Hos1p, Hos2p, Rpd31p and Rpd32p and, consistently, treatment with sodium butyrate phenocopies the decreased fluconazole resistance of both *hos2Δ/Δ and rpd31Δ/Δ C. albicans* strains [35,109].

However, the clinical success of Class I/II KDACIs in treating fungal infections is currently disputable. Valproic acid (VPA) is a branched short-chain fatty acid that acts against Class I/II KDACs [160] and is the only KDAC inhibitor of this type that has been tested in vivo. Historically VPA has been used as an anti-psychotic drug, but its anti-fungal properties have been realized as a mono-therapy against *C. albicans* [161]. More recently this property has been extended to fungi that cause mycoses of the central nervous system [162,163]. VPA’s anti-candida effect holds true for multiple NCAC species, including resistant isolates [164]. Furthermore, VPA inhibits *C. albicans* biofilm viability and interestingly this action, as well as VPA’s fungistatic effect, is pH sensitive and optimized at lower pH values [164]. Such sensitivity supports a potential application of VPA to treat vulvo-vaginal candidiasis, which persists in the acidic environment of the human vagina. VPA has also been tested in combination with amphotericin B in vitro, and this drug combination decreased biofilm viability in a dose dependent manner for *C. albicans* and *C. krusei,* but not *C. parapsilosis* [159]. Unfortunately, however, the efficacy of VPA in vitro was not recapitulated in vivo. A 2011 study found that the injection of high doses (200 mg/kg) of VPA almost doubled murine mortality in a disseminated *C. albicans* infection model, evidently by the down-regulation of host innate immunity defense genes [165]. This highlights the potential side effects of KDACIs in vivo and underscores the need to develop fungal specific inhibitors to minimize these (see below).

### 5.2. Sirtuin KDAC Inhibition in Candida spp.

Sirtuins represent Class III KDACs and for the majority of *Candida* spp. include the proteins Sir2p, Hst1p, Hst2p, Hst3p, and Hst4p. Sirtuins rely on nicotinamide adenine dinucleotide (NAD+) for deacetylation, and as such, nicotinamide (NAM) is a non-competitive inhibitor of their catalytic activity [166]. In 2010, Wurtele et al. demonstrated that 50 mM NAM is toxic to *C. albicans* and C. *krusei* and inhibits the growth of both *C. glabrata and C. parapsilosis* at higher concentrations [37]. NAM toxicity was elegantly shown to be mediated by the inhibition of Ca_Hst3p and a rise in Histone H3K56ac [37]. Furthermore, NAM used in vivo was shown to significantly drop the kidney fungal load of *C. albicans*-infected mice [37].

A more recent study has demonstrated synergistic anti-fungal effects between NAM and fluconazole in both *C. albicans* and NCAC species, including the intrinsically azole resistant *C. glabrata* and fluconazole resistant *C. albicans* isolates [167]. This study also corroborated the earlier in vivo experiments illustrating that NAM treatment results in less kidney damage, and increased survival of mice infected with *C. albicans* in a dose-dependent fashion [167]. Additionally, NAM interferes with *C. albicans* biofilm growth [167] and induces white-to-opaque switching in a manner dependent on the H3K56 acetyl-transferase RTT109 [132]. Taken together it is clear that the inhibition of sirtuins, most notably Hst3, compromises the viability of *Candida* spp., impinges on phenotypic plasticity and reverses anti-fungal resistance in these species, warranting further therapeutic-based investigation.

### 5.3. Fungal Specific KDAC Inhibitors

To date only one small molecule inhibitor has been designed that selectively targets a fungal KDAC. MGCD290 inhibits the fungal Class II KDAC, Hos2p, and was shown to have in vitro synergistic anti-fungal activity with azoles and echinocandins for *C. albicans,* NCAC species, and non-Candida fungal pathogens [168,169]. The largest chemo-sensitizing interaction observed was between MGCD290 and fluconazole, and this effect was seen even for certain fluconazole resistant *Candida* spp. isolates. A subsequent in vivo analysis showed that MGCD290 synergized with fluconazole in an invasive candidiasis (*C. albicans*) model, and mice treated with both drugs showed a significantly lower kidney fungal load and increased survival than those receiving fluconazole alone [170]. In spite of these promising results, a 2013 report on Phase II clinical trials of MGCD290 in conjunction with fluconazole to treat vulvo-vaginal candidiasis showed modest results and no significant improvement in comparison to using fluconazole alone [171]. 

As evident from these combined examples, overall KDACIs show promising therapeutic applications in treating *Candida* infections. Primarily KDACIs can be used directly to combat anti-fungal resistance by effectively returning resistance isolates to more susceptible states for numerous classes of anti-fungals. However, consistent with the potential roles of chromatin in anti-fungal resistance, KDACIs also interfere with the phenotypic plasticity of *Candida* spp. by inhibiting biofilm formation and the generation of drug tolerant persister cells. Furthermore, the observation that both TSA and SAHA-like KDACI prevent the initial adaptation of *Candida* spp. to anti-fungal drugs suggests a role for histone acetylation in the evolution and acquisition of anti-fungal resistance as illustrated in Figure 2.

However, much needs to be understood about the molecular mechanisms underlying these KDACI affects. Given the poor performance of current KDACIs in vivo it is clear that a more detailed dissection of the effects of KDACI on the *Candida* genome, transcriptome and proteome must be undertaken. This will help delineate the pertinent targets of each KDACI that give rise to the favorable outcomes seen in vitro, which will inform the design of more specific inhibitors to use in vivo. Current studies are confounded by two sub-optimal characteristics of the majority of KDACIs currently under investigation: 1. they can cross-react with host KDACs; and 2. they are pan inhibitors targeting multiple KDACs within the fungal and host cell. Taken together this means that the probability that in vivo outcomes on fungal survival and AFR will mirror what is seen in vitro is low. Going forward the community needs to focus efforts on not only fungal specific, but also protein specific KDACIs, to maximize the therapeutic potential of these drugs.

### 5.4. Mechanism of Action of KDACI on Anti-fungal Resistance—Direct or Indirect?

How do KDACIs alter anti-fungal resistance? What is the basis for the apparent chemo-sensitizing interaction between KDACIs and anti-fungal drugs? A limited number of studies have started to unravel this phenomenon mechanistically. Theoretically, three broad explanations are plausible: 1. the KDACI directly potentiates the mode of action of the anti-fungal drug; 2. the KDACI interferes with the specific anti-fungal/stress response pathways to limit the ability of the yeast cell to survive in its presence; or 3. the KDACI acts in an unrelated pathway to weaken the fungal cell and render it more vulnerable to the action of anti-fungals. Results from multiple studies are consistent with KDACIs potentiating anti-fungal drugs. For example, such a direct interaction would predict that drug synergy is very specific, and only certain combinations of KDACI and anti-fungal drug should be effective. This is true for TSA which can increase the susceptibility of *C. albicans* specifically to azoles but not echinocandins, amphotericin B or 5-flucytosine [147]. Similarly, sodium butyrate enhances the anti-fungal activity specifically of fluconazole and not of amphotericin nor caspofungin [158]. However, evidence also exists to suggest that KDACIs influence resistance mechanisms, as treatment with TSA in the presence of azoles, for example, impairs the expression of resistance genes such as *Ca_ERG1, Ca_ERG11* and *Ca_CDR1* in *C. albicans* [147]. The most convincing evidence that KDACIs act on anti-fungal resistance mechanisms is that between isolates of a given species KDACI can have contrasting effects. For example, TSA will decrease azole resistance in certain clinical isolates of *C. albicans* and not others [35], suggesting that the unique mutational pathway of a given resistant strain dictates the efficacy of this KDACI. Similarly, one of the SAHA-like KDACIs reduced fluconazole trailing growth of a specific *C. albicans* isolate, and had no effect on other isolates [148]. However, without sufficient sequencing information on such isolates, as well as clear knowledge of the pertinent molecular targets of individual KDACIs, it is difficult to make any definite mechanistic conclusions. For example, it would be interesting to ask which resistance mechanisms correlate with differences in efficacy of ClassI/II KDACIs between *C. albicans* isolates.

Conversely, other studies support an indirect effect of KDACIs, and suggest they target pathways unrelated to AFR. For example, treatment with VPA alters the *C. albicans* vacuole leading to the fungistatic nature of this KDACI [164]. Although it is known that VPA selectively targets HDAC2 in human cells [172], this remains to be confirmed for the homologous *Candida* proteins, Hos2p and Hos3p. Furthermore, it is unknown whether there is a connection between KDAC inhibition and vacuole disruption, or whether VPA has an alternative target in fungi unrelated to histone deacetylase activity. Regardless, it is pertinent to remember that not all KDACs target histone-related processes [173] and the treatment with KDACIs may have other, unrelated effects. Furthermore, many of the experiments to delineate between direct and indirect modes of action of KDACIs are complicated by the use of broad-spectrum inhibitors.

### 5.5. Anti-Chromatin Combination Therapy to Combat Candida spp. Resistance: Future Perspectives

There are noteworthy advantages to further developing anti-chromatin therapies to use synergistically with current anti-fungals to tackle the threat of *Candida* spp. resistance. For one, fungal chromatin modifiers, such as KDACs, are sufficiently diverged from their human homologues [149], providing essential avenues for exploitation in order to minimize host toxicity. Secondly, targeting non-essential proteins, such as the majority of KDACs, decreases the selective pressure on the fungal cell to obtain resistance mutations in order to alleviate inhibition of the chromatin-related target. Furthermore, combination therapy by nature increases the complexity of resistance mechanisms needed for a cell to overcome the effects of both drugs. Indeed, the synergistic interaction between a drug with limited anti-fungal effects (such as a KDACI) with a standard anti-fungal drug, so-called “syncretic interaction”, may even select against resistance, as seen for analogous antibiotic combination therapies [174,175]. Therefore, it is perhaps not surprising that adjuvant therapy for anti-fungal treatment is not unique to KDACIs [176,177]. Furthermore, this strategy offers a mechanism to extend the life of existing anti-fungal drugs, which given the challenges in developing novel therapeutics is both highly desirable and cost-effective. Lastly, many KDACIs are already pre-approved for human use, potentially shortening the pipeline from discovery to clinic for fungal specific KDACIs.

Presented here is a model for the role of epigenetically regulated phenotypic resistance in the acquisition of AFR. The possibility that this evolutionary process can be delayed through the use of chromatin modification modulators, such as KDACIs, is attractive and initial studies show promise for this pharmacological approach. Such ‘anti-evolution’ therapy has been explored to varying levels of success in the manipulation of biological systems including cancer, and numerous strategies exist [178,179]. Targeting epigenetic mechanisms, however, leads to some unique considerations. For one, reprogramming the epigenome requires a long-lasting effect and therefore the efficacy of a drug will depend on the extent that specific histone modification states are inherited mitotically. Secondly, histone modifiers act at many different loci and, as is evident from the genetic studies, removal of a single histone modifier can be pleiotropic and, in some instances, lead to conflicting phenotypes. Thirdly, the flexibility of the epigenome may mean that cells gain tolerance to potential anti-chromatin therapy easily, rendering any drug ineffective over time. However, identifying the key time points to administer any potential anti-evolution therapy might maximize its effect and maintain sensitivity to the drug. For example, intermittently delivering KDACI during a course of anti-fungal treatment might show the greatest benefit, by blocking the cells from exploring epigenetic space under stress and gaining phenotypic resistance. Valid in vivo models of candidiasis, using strains with varying degrees of phenotypic resistance, will undoubtedly help inform such therapeutic strategies.

However, we remain far from actualizing the therapeutic potential of KDACIs currently. Although there is substantial genetic data to support anti-chromatin therapy for *Candida* infections, the field suffers from a lack of information on the biochemistry of *Candida* spp. proteins. From the point of view of KDACs, to date only *C. albicans* Hos2p has been purified and assessed in vitro and, unexpectedly, Hos2p is inactive on acetylated histones but capable of deacetylating tubulin [180]. Therefore, it could be hypothesized that Hos2p inhibition by MGCD290 may be indirectly compromising the fungal cytoskeleton, and not influencing drug resistance mechanisms. However, verification of the substrate preference of Hos2p in vivo is still required to provide traction for this hypothesis. Needless to say, substrate specificity of potential KDACI targets needs to be undertaken, as well as mechanistic studies, such as chromatin immunoprecipitation followed by sequencing (ChIPseq), to elucidate the precise mechanistic details that explain the phenotypes observed with various KDAC mutant strains. Most importantly, structure-function information about *Candida* KDACs is lacking that would aid in the design of fungal-specific inhibitors. Furthermore, biochemical analysis is required to further the development of robust and high throughput assays necessary to screen potential yeast-specific inhibitors of chromatin modifiers, such as the one outlined for *C. albicans* Hos2p function [180].

## 6. Concluding Remarks

Undoubtedly adaptation to anti-fungal drugs and the evolution of AFR is primarily a genetic process; however, chromatin structure can influence not only the trajectory of genetic adaptation, but also its phenotypic expression. We have presented a framework to think about how epigenetic mechanisms might facilitate AFR through an initial phase of phenotypic resistance, that provides for a small fraction of cells within a population an increased opportunity to acquire genetic resistance. We also include in our definition of phenotypic resistance the ability of *Candida* cells to switch to biofilm growth as well as distinct cellular morphologies, both strategies of which alter anti-fungal susceptibilities. This demonstrates that phenotypic plasticity, common to pathogenic *Candida* species, is an advantageous strategy to survive anti-fungal stress.

Genetic experiments have contributed substantially to our understanding of how chromatin modifiers influence phenotypic resistance especially in *C. albicans*, but now there is a need to extend efforts to NCAC species, most notably, the multi-drug resistant *C. auris*. Furthermore, there are strong arguments to be made for developing modulators of epigenetic machinery to tackle AFR, and KDACIs are one class of drugs that show promising therapeutic effects. However, further mechanistic studies on *Candida* spp. phenotypic heterogeneity, and how these states are dictated by specific chromatin modifiers, are needed. Such studies will undoubtedly generate further therapeutic targets to combat phenotypic resistance, and potentially delay the development of anti-fungal resistance.

## Figures and Tables

**Figure 1 jof-06-00121-f001:**
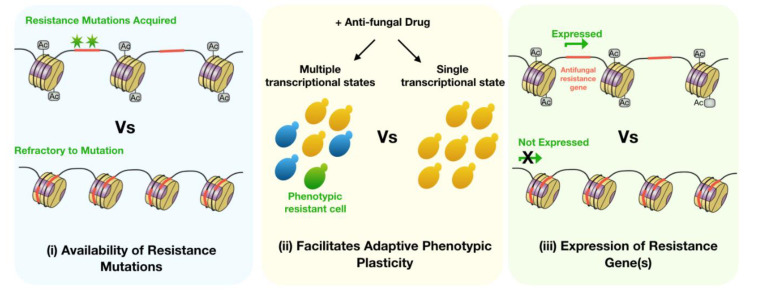
Potential mechanisms for how chromatin state influences anti-fungal resistance in *Candida* spp. (**i**) Differences in the accessibility of DNA throughout the genome, dictated by chromatin state, can impact where genetic variation may occur. This means that chromatin may potentially control the availability of specific anti-fungal resistance mutations (green stars). (**ii**) Phenotypic plasticity allows certain *Candida* ssp. to display altered phenotypes in response to environmental cues and gives rise to population heterogeneity (indicated by different coloured yeast cells). Phenotypic plasticity relies on the relaxation of barriers that regulate transcription, including chromatin structure, and can result in a transcriptional state that facilitates phenotypic resistance (green yeast cell). (**iii**) Chromatin structure directly regulates transcription and therefore can influence whether a given resistance mutation is expressed or not. Furthermore, certain anti-fungal resistance mechanisms rely on the overproduction of a normal gene, which can be sensitive to particular chromatin state(s). (Ac = acetyl group).

**Figure 2 jof-06-00121-f002:**
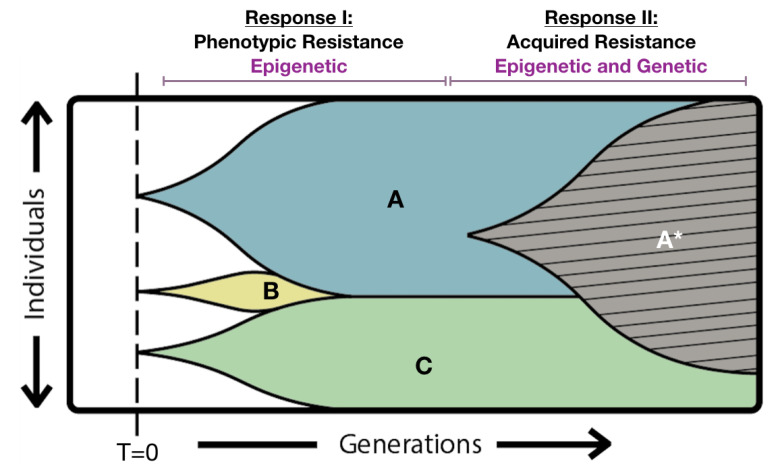
Theoretical epigenetic and genetic combination model to explain *Candida* spp. growth in the presence of high concentrations of anti-fungal drug. The diagram shows the epigenotype and genotype of an evolving population of *Candida* yeast cells when exposed to an antifungal drug (concentration > MIC) at time 0 (T = 0). Three individual cells are capable of growing under these conditions and their descendants give rise to sub-populations A, B, and C. We postulate that this ability to grow is dictated by an advantageous epigenetic program within each of these cells either at the time of antifungal exposure, or very soon thereafter. We refer to this as response I, which accounts for the trailing growth phenotype typically seen 48 h after drug exposure. In this particular example, sub-population B is outcompeted by sub-populations A and C which presumably have greater relative adaptive fitness. After some time, a cell in sub-population A acquires a genetic lesion that confers anti-fungal resistance (A*), providing a selective advantage over its non-mutated ancestors, and the population as a whole has entered phase II, the acquired resistance phase. In the time period depicted, a fraction of descendants from the epigenetically defined, phenotypically resistant population C remain.

**Figure 3 jof-06-00121-f003:**
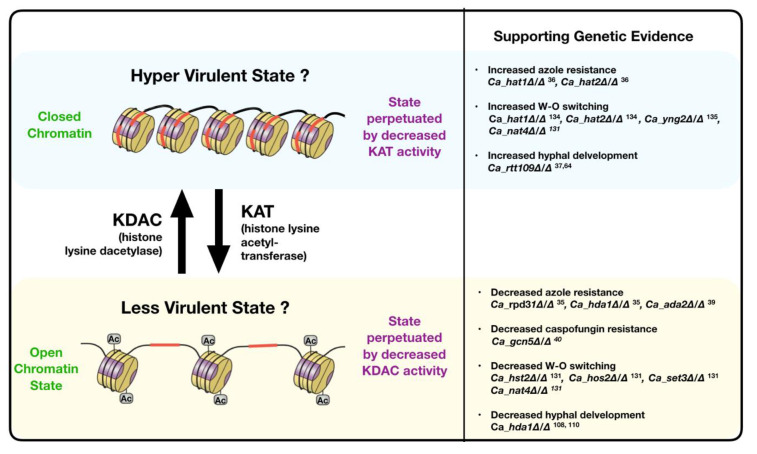
Simplified model of the role of histone acetylation in *C. albicans* virulence. A summary of the genetic studies in *C. albicans* that are consistent with a model whereby a decrease/loss in specific histone acetylation marks lead to a hypervirulent state, whereas an increase in histone acetylation is associated with the attenuation of virulence. All examples are from studies in *C. albicans* specifically. This model is the rationale for why (K) deacetylase (KDAC) inhibitors are potentially useful anti-fungal drugs. (Ac = acetyl group).

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
