# Peer review of "Chromatin Structure and Drug Resistance in Candida spp."

_jof, 2020, doi:10.3390/jof6030121_

Round 1

Reviewer 1 Report

Line 2 – The title does not fit to the clue of manuscript, that is anti-chromatin therapy

Lines 8-21 - The summary does not reflect the balance between the informations in the manuscript. There is written only one sentence in summary („We also discuss the therapeutic potential of targeting Candida spp. chromatin modifiers in the fight against AFR.”) on the topic dissccused in details in lines 446-664”

Line 31 - citation is not adequate

Line 33 - double spaced

Line 39 - there is :anti-fugal" instead of "anti-fungal"

Line 46 – lack of the fifth group of antifungals:  allylamine (e.g. terbinafine) inhibits the action of squalene epoxidase

Line 59 – the citation is not recent

Line 115 – List of the three chromatin-mediated mechanisms does not fit to flow of the text

Lines 121-123 – „In this review we will focus our discussion on possibilities 1 and 2, and direct readers to other reviews that extensively describe the transcriptional regulation of anti-fungal resistance mechanisms (39, 40)” – It is not clear why only 2 citations are presented while the issue is „extensively described”

Line 180 – There is „have not be” instead „have not been”

Line 197 – double spaced

Line 272 – The paragraph 4.2 is too general – there is lack of description of molecular mechanisms

Line 282 – There is „exist” instead of „exists”

Line 324 – „we can tackle biofilm resistance through the manipulation of the 324 histone de/acetylation pathways involved in Y-H transition.: - expression „we can tackle” is not correct. Please, check in the manuscript the sentences starting with „We” (lines 323 - 337) – consider if „we” is overused

Line 393 – Citation 137 is not adequate

Line 453 – „(and Hyland lab 453 unpublished)” should be removed

Line 467 – „(and Hyland lab 453 unpublished)” should be removed

Line 473 – The Figure 3 is of great importance. Could Authors change it to make it more „reader friemdly”?

Line 478 – double spaced

Line 506 –Abberivation „NaB” can not be used for Natrium butyrate, as it means sodium benzoate

Line 523 – There is „Volvo-vaginale candidassi” instead „vulvovaginal candidiasis”

Lines 633 – 664 – Avoid „we”

Lines 685… - References are double numbered

Reviewer 2 Report

O´Kane and co-workers present the review entitled "Chromatin structure and drug resistance in Candida spp.", focused in the genetic mechanisms that can influence and dictate antifungal resistance. Given the increment in fungal resistance to the drugs actually marketed, the topic of the review is pertinent and actual. The work is very well written and organized, and the authors have focused the most important points, giving a comprehensive point of view of the role of chromatin structure in fungal resistance.

I enjoyed reading this work, and I believe it would be of interest for the readers of JoF.

Author Response

Nothing to address.